# Hog1 acts in a Mec1-independent manner to counteract oxidative stress following telomerase inactivation in *Saccharomyces cerevisiae*
Bechara Zeinoun, Maria Teresa Teixeira ✉ & Aurélia Barascu ✉

Replicative senescence is triggered when telomeres reach critically short length and activate permanent DNA damage checkpoint-dependent cell cycle arrest. Mitochondrial dysfunction and increase in oxidative stress are both features of replicative senescence in mammalian cells. However, how reactive oxygen species levels are controlled during senescence is elusive. Here, we show that reactive oxygen species levels increase in the telomerase-negative cells of *Saccharomyces cerevisiae* during replicative senescence, and that this coincides with the activation of Hog1, a mammalian p38 MAPK ortholog. Hog1 counteracts increased ROS levels during replicative senescence. While Hog1 deletion accelerates replicative senescence, we found this could stem from a reduced cell viability prior to telomerase inactivation. ROS levels also increase upon telomerase inactivation when Mec1, the yeast ortholog of ATR, is mutated, suggesting that oxidative stress is not simply a consequence of DNA damage checkpoint activation in budding yeast. We speculate that oxidative stress is a conserved hallmark of telomerase-negative eukaryote cells, and that its sources and consequences can be dissected in *S. cerevisiae*.

Telomeres are essential structures found at the ends of linear eukaryotic chromosomes, consisting of DNA sequences, proteins, and long non-coding RNA (LncRNA) telomeric transcripts[1]. Telomeres crucially safeguard chromosome integrity by protecting against degradation and fusion events[2]. However, due to the "DNA end-replication problem", telomeres gradually shorten with each cell cycle. Telomerase, a specialized reverse transcriptase, counteracts telomere shortening by adding repetitive telomeric sequences to chromosome ends. In human somatic cells, the telomere-protective functions become compromised when the telomeres shorten with cell divisions due to telomerase inactivation coupled with the "DNA end-replication problem". When telomere lengths become critically short, they activate an irreversible DNA damage checkpoint-dependent cell cycle arrest, known as replicative senescence[3,4]. The unicellular eukaryote, *Saccharomyces cerevisiae*, relies on telomerase for its long-term viability[5], but similar to human somatic cells, telomerase inactivation in budding yeast also leads to replicative senescence. When budding yeast cells divide in the absence of telomerase, they cease proliferation and enter a metabolically active state, arresting in the G2/M phase of the cell cycle[6,7]. This cell cycle arrest in *S. cerevisiae*, which is akin to mammalian cells, relies on the activation of the DNA damage checkpoint kinases, Mec1 and Tel1 (the yeast orthologs of ATR and ATM, respectively), in addition to Rad53 phosphorylation[8–10]. Remarkably, not only is the triggering of replicative senescence in response to short telomeres evolutionarily conserved, but also many other essential telomeric functions and maintenance mechanisms[11,12].

Studying replicative senescence is challenging due to the inherent heterogeneity resulting from intracellular differences in telomere lengths and the immense intercellular variations[13]. Intriguingly, data collected from various organisms indicate that mitochondrial defects, oxidative stress, and chronic inflammation can accelerate telomere shortening and dysfunction[14–17]. These factors are potential sources of cell-to-cell variation and contribute to genome instability during replicative senescence[18]. Notably, senescent human fibroblasts exhibit modifications in mitochondrial structure and function, accompanied by elevated reactive oxygen species (ROS) levels and oxidative damage[19–24]. Similar metabolic alterations have also been observed in budding yeast during replicative senescence. A previous study revealed that the absence of telomerase resulted in increased mitochondrial mass, and a transcriptomic analysis indicated that energy production genes were up-regulated and stress response genes were

Sorbonne Université, PSL, CNRS, UMR8226, Institut de Biologie Physico-Chimique, Laboratoire de Biologie Moléculaire et Cellulaire des Eucaryotes, F-75005 Paris, France. ✉e-mail: teresa.teixeira@ibpc.fr; aurelia.barascu@ibpc.fr

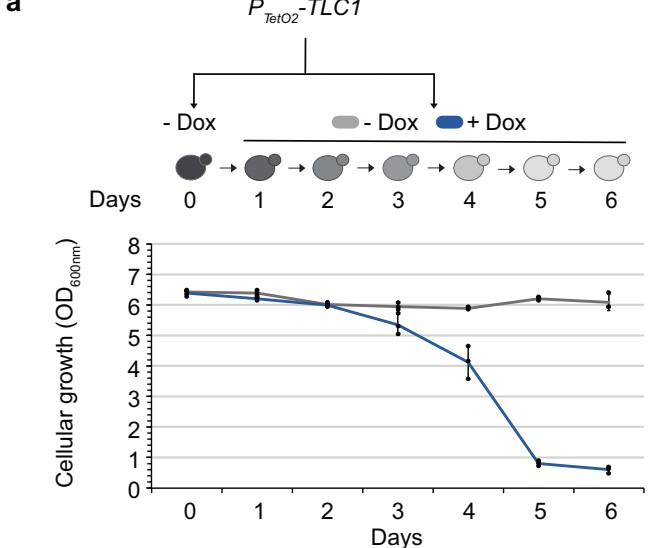

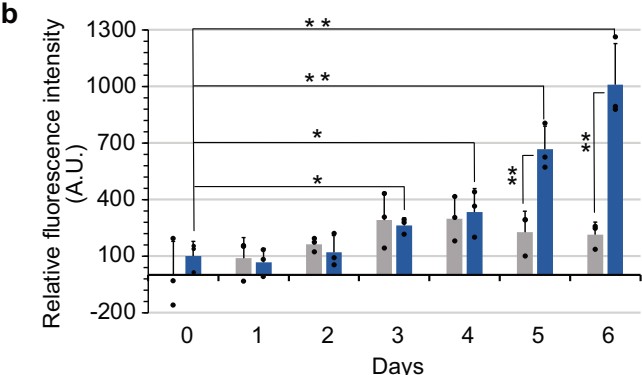

**Fig. 1 | ROS levels increase during replicative senescence in budding yeast.** Each consecutive day, cells with the genotype indicated were diluted in media either with or without doxycycline (Dox) to enable conditional shut-off of telomerase, and grown for 24 h. Cell density at OD$_{600nm}$ (**a**) and ROS levels normalized to the strain without doxycycline (**b**) are plotted as mean ± SD of three independent clones. *P* values were calculated by two-tailed Student's *t* test (* < 0.05; ** < 0.01) and only significant differences have been represented.

induced[25]. However, data regarding ROS level alterations and their regulation during senescence in budding yeast is currently lacking.

P38, a member of the mitogen-activated protein kinase (MAPK) family, is critical for various cellular processes, including cellular senescence and oxidative stress responses[26]. In budding yeast, the MAPK Hog1, the ortholog of mammalian p38, is crucial for the defence against many stressors, including osmotic[27] and oxidative stress[28]. The canonical pathway of Hog1 activation involves two branches, the Sho1 and Sln1 branches, which converge to activate the MAPK kinase (MAPKK), Pbs2[29]. Pbs2 then interacts with and phosphorylates Hog1 at the conserved residues, Thr[174] and Tyr[176], leading to its activation. Hog1 is a multifunctional protein with important functions in both the cytoplasm and nucleus, and it is vital for stress adaptation[30]. Its roles encompass regulating gene expression by activating transcription factors, participating in gene initiation and elongation, regulating the cell cycle, and contributing to various steps in mRNA metabolism. Notably, Hog1 is activated in response to H$_2$O$_2$ stress[28] and regulates antioxidant genes by activating the transcription factors, Msn2/Msn4[31] and Sko1[32]. In the absence of Hog1, cells become more sensitive to H$_2$O$_2$, which was previously shown to correlate with reduced expression of the *TSA2* gene[31]. Conversely, sustained Hog1 activation can lead to cell death, which has been linked to alterations in mitochondrial respiration and

increases in ROS levels[33]. Hog1 counters this ROS increase by inducing *PNC1* and activating Sir2. Multiple studies have demonstrated that uncontrolled Hog1 activation disrupts mitochondrial function and elevates ROS levels, underscoring the critical importance of regulating Hog1 activation[34,35]. Furthermore, Hog1 has been implicated in autophagic processes and is required for mitophagy, the selective process of mitochondria degradation[36–38]. Interestingly, Hog1 also positively regulates the localization of the Sir complex to telomeres following osmotic stress and the silencing of telomeric regions[39].

Here, we report that ROS levels increase during replicative senescence in budding yeast. During this process, Hog1 is activated by Pbs2 and plays a role in ROS detoxification. This countering of ROS increase occurs independently from the actions of Mec1. We also find that autophagy does not participate in replicative senescence in budding yeast. However, Hog1 participates in maintaining telomere length homeostasis and affects cell viability. Our results thus suggest that Hog1 serves as a link between telomeres and ROS metabolism.

## Results

### ROS levels increase during replicative senescence in budding yeast

Human senescent cells exhibit increased ROS levels during replicative senescence, however, data relating to *S. cerevisiae* senescent cells is lacking[19,20]. Budding yeast telomerase is constitutively active but experimentally inactivating it triggers replicative senescence[8]. We used a validated *TetO2-TLC1* system, where *TLC1*, which encodes the telomerase RNA template, is controlled by a repressible promotor by doxycycline, enabling the conditional shut-off of telomerase[10,40,41]. To measure ROS levels, we used DCF, a molecule that can be directly oxidized by ROS and produce fluorescence in quantities reflecting ROS levels, which we can quantify by flow cytometry. Replicative senescence was observed in the P$_{TetO2}$-*TLC1* strain starting from day three of culture with doxycycline, as the cell proliferation capacity gradually decreased, eventually reaching a crisis point by day 5 (Fig. 1a). We also observed a simultaneous increase in ROS levels in the absence of telomerase all along the experiment in presence of doxycycline with a significant increase as compare to the untreated condition starting from the day 5 (Fig. 1b). These results were recapitulated in strains where *TLC1* was deleted; in these strains, we also found that ROS levels declined as cultures recovered their initial proliferation capacity following the emergence of post-senescence survivors (Supplementary Fig. 1). As previously outlined, during prolonged culturing of senescent cells, rare events, estimated to occur at a frequency of approximately $2 \times 10^{-5}$ cells, enables cells to circumvent senescence and resume cellular divisions[42,43]. These dividing cells, referred to as post-survivors, have the ability to replenish the culture and sustain their telomere length independently of telomerase activity. Instead, they rely on recombination mechanisms, known as the alternative lengthening of telomeres mechanism, which are conserved in yeast and mammals. These data indicate that similar to mammalian models, budding yeast telomerase-negative senescent cultures exhibit increased ROS levels.

### Hog1 is activated during replicative senescence in a Pbs2-dependent manner and counteracts increase in ROS levels

We next investigated how ROS are regulated during replicative senescence. We focused on the multifunctional MAPK, Hog1, which is activated by oxidative stress[28,30]. To determine whether Hog1 activation occurred during replicative senescence, we prepared protein extracts from senescent cultures and used a specific antibody to detect phosphorylated forms of Hog1. We observed that Hog1 was phosphorylated during replicative senescence from day three, concomitant with an increase in ROS levels (Fig. 2d or Supplementary Fig. 2b and Fig. 2b, respectively). The MAPKK Pbs2 precedes Hog1 in its canonical pathway and is the only Kinase capable of phosphorylating Hog1[29]. Therefore, the Western blot presented in Fig. 2d, showing that the phosphorylated form of Hog1 is no longer detectable in the *PBS2*-deleted mutant, confirms the deletion. We also observed that *HOG1* or *PBS2* deletion (P$_{TetO2}$-*TLC1 hog1Δ* or P$_{TetO2}$-*TLC1 pbs2Δ*) (Fig. 2c and

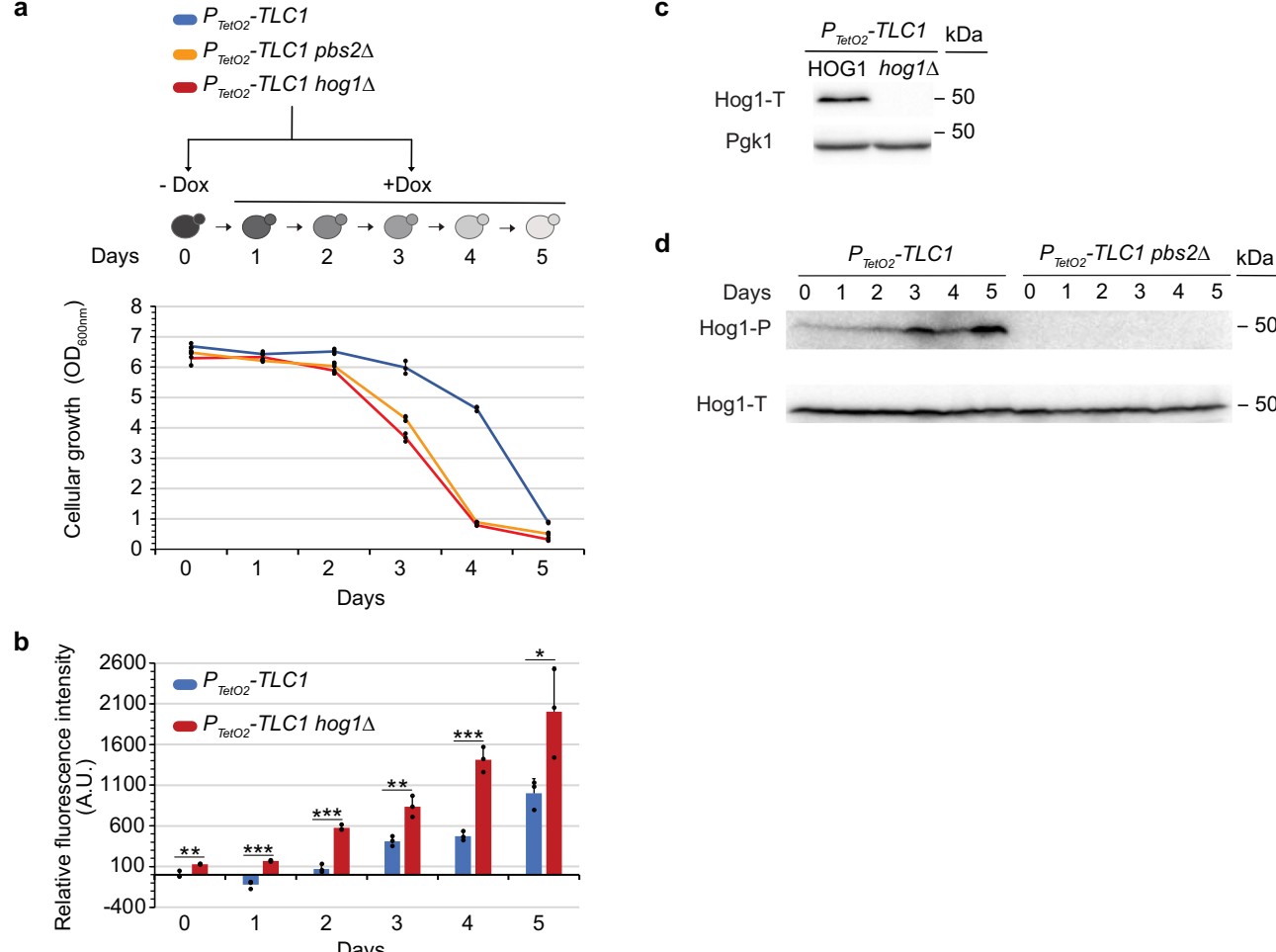

**Fig. 2 | Hog1 is activated during replicative senescence in a Pbs2-dependent manner and counteracts increase in ROS levels.** Cells with the genotypes indicated were treated (D1 to D5) or not (D0) with doxycycline as described in Fig. 1. Cell density at OD600nm (**a**) and ROS levels normalized to the $P_{TetO2}$-TLC1 strain without doxycycline (**b**) are plotted as mean ± SD of three independent clones. *P* values were calculated by two-tailed Student's *t* test (* < 0.05; ** < 0.01; *** < 0.001) and only significant differences between the two strains used have been represented. **c**, **d** Protein extracts, from experiment presented in (**a**), analysed by Western blot using an antibody against phosphorylated forms of Hog1's human ortholog p38 (Hog1-P), total Hog1 (Hog1-T) or against Pgk1.

Supplementary Fig. 2a or Fig. 2d, respectively) resulted in a similar premature loss of cell viability, when telomerase was inactivated (Fig. 2a). This suggests that the Hog1 pathway may play a role in senescent cells. While the HOG1 pathway activates transcription factors of antioxidant genes to reduce ROS levels[33], excessive Hog1 activity can increase ROS levels by disrupting mitochondrial respiration[33,34]. To understand which of these Hog1 functions was involved in replicative senescence, we measured ROS levels in the absence or presence of Hog1 during senescence. Our results showed that the increases in ROS levels started earlier and reached higher levels in $P_{TetO2}$-TLC1 *hog1Δ compared* to $P_{TetO2}$-TLC1 strains (Fig. 2b). We also observed that *HOG1* or *PBS2* deletion showed a similar increase in the level of ROS in the course of senescence (Supplementary Fig. 3), confirming the epistatic interaction between the two genes. This suggests that increases in ROS levels trigger an oxidative stress response that activates *via* Pbs2 the Hog1 pathway, which is required for ROS detoxification in telomerase-inactivated cells.

### *HOG1* deletion affects telomere length homeostasis and cell viability

Given the direct relationship between replicative senescence and telomere shortening, we investigated the potential influence of Hog1 on telomere length homeostasis prior to telomerase inactivation, and the rate of telomere shortening in the absence of telomerase. We thus performed telomere-PCR on DNA samples from $P_{TetO2}$-TLC1 and $P_{TetO2}$-TLC1 *hog1Δ* strains to determine telomere length. Our results showed that *HOG1* deletion resulted in slightly shorter telomeres of ~30 bp prior to telomerase inactivation (Supplementary Fig. 5a, b). However, no significant differences in telomere shortening rates were observed in the absence of telomerase in either the $P_{TetO2}$-TLC1 or $P_{TetO2}$-TLC1 *hog1Δ* strains; shortening rates were measured to be approximately 2.5 bp/cell population doubling (Fig. 3a, b, Supplementary Fig. 4), similar to previously published results[40,44]. Therefore, one plausible hypothesis is that Hog1 contributes to the maintenance of telomere length homeostasis. As telomere length homeostasis results from the balance between telomere lengthening by telomerase and telomere shortening due to "DNA end replication problem", we suggest that Hog1 might promote telomerase recruitment or activity.

To test if the accelerated senescence we observed in the *hog1Δ* strains, when telomerase was inactivated, is due to shorter telomere lengths prior to telomerase inactivation, we evaluated the senescence curve of the *TLC1 HOG1* (the wild type condition), *TLC1 hog1Δ*, $P_{TetO2}$-TLC1 and $P_{TetO2}$-TLC1 *hog1Δ* haploids cells from tetrad dissection. We can observe that the change from the endogenous TLC1 promoter to the $P_{TetO2}$ promoter induces a shortening of telomere length, as previously characterized[10] (Supplementary Fig. 5c, d). Moreover, significant clonal variability in telomere length was previously reported[45]. We find this variability among the majority of clones exhibiting the same genotype (Supplementary Fig. 5c, d).

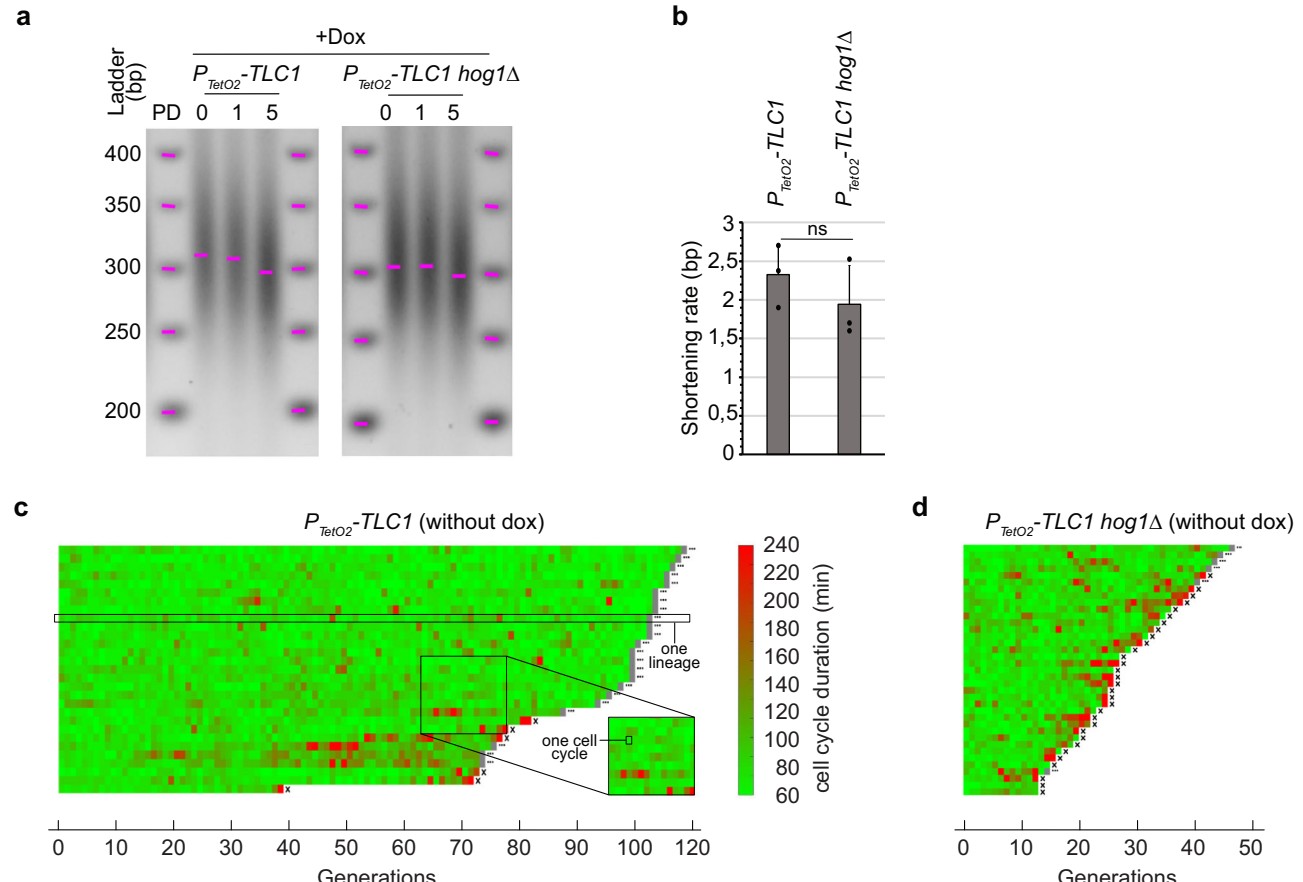

**Fig. 3 | Impact of HOG1 deletion on telomere shortening and cell viability. a** Cells of the genotypes indicated were pre-cultured overnight in doxycycline-containing media to enable telomerase shut-off. Cells were then diluted and grown in the same media for the indicated population doublings (PD). Representative telomere-PCR of Y' telomeres from genomic DNA extracts are shown. **b** Quantification of telomere shortening was measured between PD 1 to 5 and plotted as mean ± SD. **c, d** Microfluidics results of independent lineages with the genotypes indicated. Cells were introduced into the microfluidics microcavities and cultured in SD without doxycycline. Each horizontal line represents the consecutive cell cycles (generations) of a single lineage, and each segment corresponds to one cell cycle. An ellipsis (…) at the end of the lineage line indicates that the cell was living after the experiment, whereas an X indicates cell death. Cell cycle duration is indicated by the colored bar. Scripts used to generate Fig. 3c, d are available: https://doi.org/10.5281/zenodo. 11634847.

In this context, the small but reproducible decrease in telomere size, significantly observed in three independent $P_{TetO2}$-*TLC1 hog1Δ* transformants compared to $P_{TetO2}$-*TLC1* (Supplementary Fig. 5a, b), is no longer observed in a context of tetrad dissection (Supplementary Fig. 5c, d). Yet, the proliferation of $P_{TetO2}$-*TLC1 hog1Δ* spores in the presence of doxycycline, consistently show accelerated senescence compared to $P_{TetO2}$-*TLC1* sister spores (Supplementary Fig. 5e). Based on these results, we can conclude that the acceleration of senescence in the absence of Hog1 is independent of the initial shorter telomere length observed in this strain. As expected, the growth curves of the *HOG1* and the *hog1Δ* cells, where *TLC1* has is endogenous promoter, are not affected over five days with doxycycline (Supplementary Fig. 5e). Altogether, these data help us separate the slight decrease in telomere size observed in $P_{TetO2}$-*TLC1 hog1Δ* before telomerase inactivation from the accelerated senescence observed in the same strain after telomerase inactivation.

Another hypothesis to account for the acceleration of senescence in $P_{TetO2}$-*TLC1 hog1Δ* cells is that the deletion of *HOG1* might influence cellular mortality. To test this, we employed a microfluidics system, which allows consecutive cell cycles from individual cell lineages (herein referred to as lineages) to be tracked, thereby enabling more precise quantification of cell proliferation[46]. In the presence of telomerase, $P_{TetO2}$-*TLC1* cells grew indefinitely with a spontaneous mortality rate of ~0.38% (Fig. 3c). In contrast, the absence of *HOG1* caused the mortality rate to increase to ~5.8% even in the presence of telomerase (Fig. 3d). This has not previously been observed with cell population growth (liquid or solid). This data indicates that *HOG1* loss alone causes some cell death, which, when combined with telomerase inactivation, could have contributed to the accelerated senescence we observed.

Collectively, these findings suggest that the apparent acceleration of senescence we observed in the absence of Hog1 could be attributable to a marked increase in intrinsic mortality rates. However, this does not preclude a significant role for Hog1 in detoxifying ROS during replicative senescence, particularly as $P_{TetO2}$-*TLC1 hog1Δ* strains exhibit much higher ROS levels throughout replicative senescence compared to $P_{TetO2}$-*TLC1* strains (Fig. 2b and Supplementary Fig. 3). To understand the impact of higher basis level ROS on the viability of $P_{TetO2}$-*TLC1 hog1Δ* cells, we evaluated the cell proliferation of $P_{TetO2}$-*TLC1 hog1Δ* cells *versus* $P_{TetO2}$-*TLC1* treated with the antioxidant N-acetyl-L-cysteine (NAC) 10 mM over 10 generations in the exponential phase. The NAC treatment significantly improved proliferation and ROS levels in both strains, however $P_{TetO2}$-*TLC1 hog1Δ* showed the most improvement (Supplementary Fig. 6a, b, respectively). Interestingly, we observed that $P_{TetO2}$-*TLC1 hog1Δ* cells treated with 10 mM NAC during exponential phase exhibited a partial rescue of telomere length. However, the telomere length of $P_{TetO2}$-*TLC1* cells remained unaffected (Supplementary Fig. 6c, d). This data underscores the sensitivity of telomeric sequences to ROS compared to the rest of the genome[47], emphasizing the importance of Hog1 in maintaining ROS balance under basal condition.

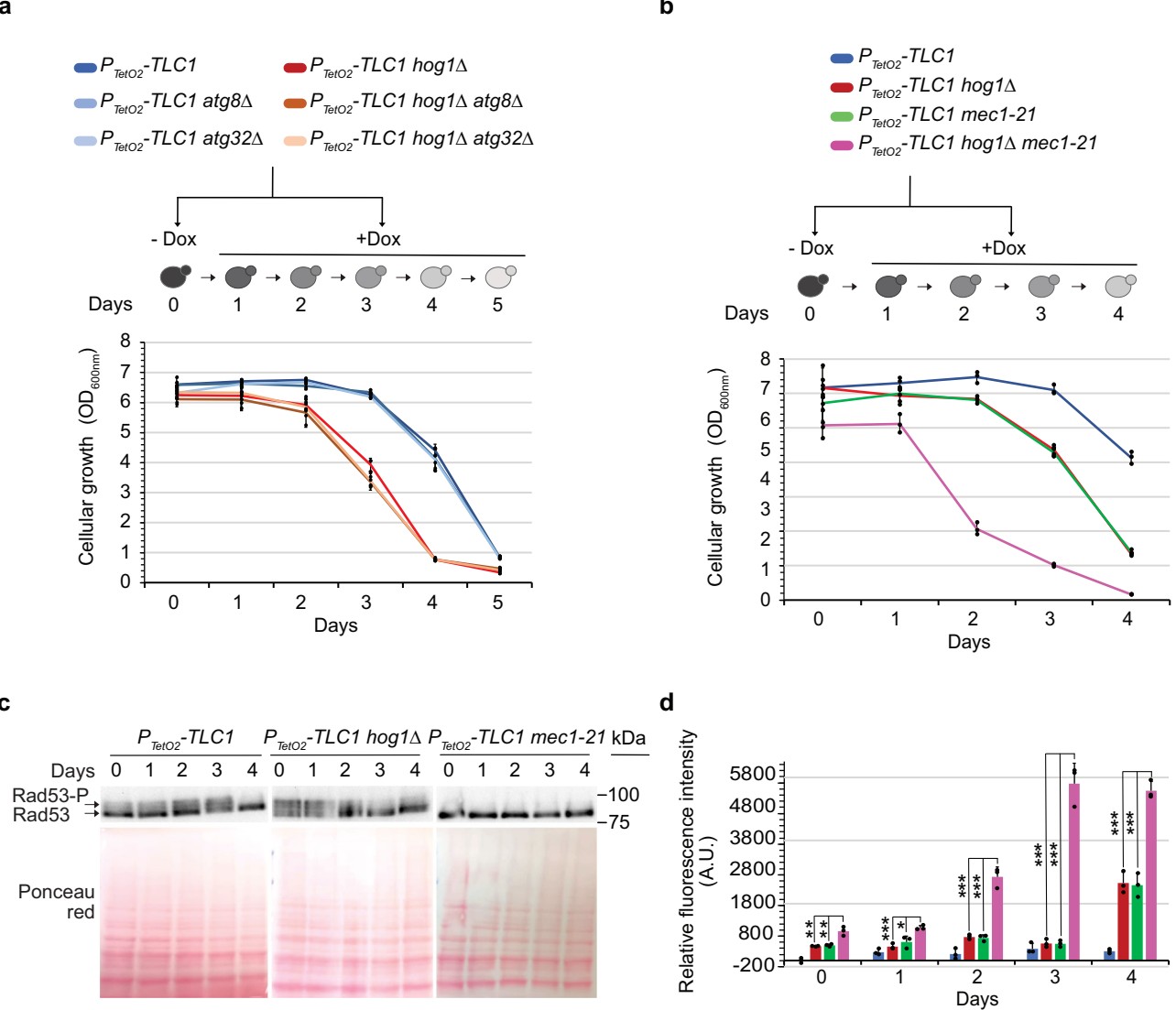

**Fig. 4 | Hog1 interaction with multiple pathways. a, b** Cells with the genotypes indicated were treated as described in Fig. 1. Cell density at $OD_{600nm}$ from the strains indicated are plotted as mean ± SD of three independent clones. **c** Results derived from the senescence experiment depicted in Fig. 4b. Protein extracts analysed by Western blot using an antibody against Rad53. The mobility shift of the band indicates Rad53 phosphorylation (the upper arrow). **d** ROS levels normalized to the $P_{TetO2}$-TLC1 strain without doxycycline derived from the senescence experiment depicted in Fig. 4b using the same legend are plotted as mean ± SD of three independent clones. P values were calculated by two-tailed Student's t test (* < 0.05; ** < 0.01; *** < 0.001) and only significant differences between the triple mutant and the double mutants have been represented.

## Autophagic processes do not modify the onset of replicative senescence

Autophagy is a process that involves self-eating and bulk degradation, where organelles and their components are delivered to vacuoles to be degraded and recycled[48]. Autophagy can be selective, when specific cargo, such as damaged organelles, are degraded[49]. The specific degradation of dysfunctional mitochondria is known as mitophagy. Hog1 plays a role in autophagy under certain conditions[50–52] and is considered to be a mitophagy activator[36,37,53]. We therefore investigated whether autophagic processes were essential during replicative senescence. We deleted the *ATG8* and *ATG32* genes, which encode two proteins essential for bulk autophagy and mitophagy in budding yeast, respectively[49]. These mutants $P_{TetO2}$-TLC1 *atg8Δ* and $P_{TetO2}$-TLC1 *atg32Δ* displayed a blockage of bulk autophagy and mitophagy respectively, confirmed by an assay based on Rosella, a fluorescence-based pH biosensor[54] (Supplementary Fig. 7).

We then investigated whether blocking these processes in wild-type cells would affect senescence dynamics. Liquid senescence assays revealed that senescence remained unchanged in the absence of autophagy or mitophagy, suggesting that these processes were not essential for the viability of telomerase-negative budding yeast cells (Fig. 4a). Similarly, following telomerase inactivation, the senescence profiles remained unchanged when either *ATG8* or *ATG32* were deleted in a *hog1Δ* background (Fig. 4a). We concluded that Hog1 activity occurs independently from autophagic processes, which do not alter the onset of replicative senescence in budding yeast.

## Hog1 acts in a Mec1-independent manner to regulate ROS levels during replicative senescence

We hypothesized that ROS level increases could result from cells being in a senescent state. Mec1 is a pivotal kinase in budding yeast, essential for the DNA damage checkpoint and the onset of replicative senescence[55]. Mec1 and the DNA damage checkpoint pathway are also known to protect cells against oxidative stress[56]. Hog1 and Mec1 are both necessary to combat the oxidative stress induced by hydrogen peroxide, but act independently[28]. A hypomorph mutant of *MEC1, mec1-21*, contains a G to A substitution at position 2644, outside the kinase domain[57]. The *mec1-21* mutant displays

## Table 1 | strains used in this study

| Strain name | Figure | Genotype |
|---|---|---|
| yT787 | 1, 2, 3, 4, S2, S4, S5, S6, S7, S9 | *Mata ura3-1 trp1-1 leu2-3,112 his3-11,15 tlc1::HIS3MX6-P$_{TetO2}$-TLC1* |
| yT1473 | 2, 3, 4, S2, S3, S4, S5, S6, S7, S9 | *yT787 hog1::TRP1* |
| yT1714 | 2, S3 | *yT787 pbs2::TRP1* |
| yT1743 | 4, S8, S9 | *yT787 mec1-21* |
| yT1744 | 4 | *yT787 hog1::TRP1 mec1-21* |
| yT1701 | 4, S7 | *yT787 atg8::NAT* |
| yT1702 | 4, S7 | *yT787 hog1::TRP1 atg8::NAT* |
| yT1705 | 4, S7 | *yT787 atg32::NAT* |
| yT1706 | 4, S7 | *yT787 hog1::TRP1 atg32::NAT* |
| yT338 | S1 | *TLC1/ tlc1Δ* |
| Diploid | S5 | *TLC1/ tlc1Δ HOG1/ hog1::TRP1* |

lower dNTP levels and shorter telomeres (~50 bp) compared to wild type strains[57,58]. The *mec1-21* mutant retains essential functions but is defective for the S phase checkpoint and Rad53 activation following UV and HU exposure[59,60]. We thus used the *mec1-21* mutant to investigate whether the actions of Hog1 against ROS were Mec1-dependent during replicative senescence. Even prior to telomerase inactivation (D0), we observed a slight decrease in proliferation in the single P$_{TetO2}$-TLC1 *mec1-21* strain indicating an increase in the proportion of deceased cells (Fig. 4b). This trend is further pronounced in the double mutant P$_{TetO2}$-TLC1 *mec1-21 hog1*. This observation remains consistent for the ROS levels observed in Fig. 4d at D0. Therefore, it is well possible that ROS levels follow mortality. Then we inactivated telomerase and measured cell growth over time in the P$_{TetO2}$-TLC1, P$_{TetO2}$-TLC1 *hog1Δ*, P$_{TetO2}$-TLC1 *mec1-21*, and triple mutant strains. When compared to *MEC1* cells, we observed that *mec1-21* accelerated the loss of viability under telomerase-negative conditions (Fig. 4b), consistent with the initial shorter telomeres (Supplementary Fig. 8a, b). We also verified that Rad53 phosphorylation was impaired in the *mec1-21* strains (Fig. 4c and Supplementary Fig. 9). Yet, under these conditions, where the DNA damage checkpoint was disabled, telomerase inactivation resulted in more pronounced increase in ROS levels. This indicates that Mec1 also participates in ROS detoxification in the absence of telomerase (Fig. 4d). In addition, while the triple mutant, P$_{TetO2}$-TLC1 *hog1Δ mec1-21*, displayed a much lower proliferation capacity, it showed an even more pronounced increase in ROS compared to the respective single mutants (Fig. 4d). As we showed in Supplementary Fig. 6a, b, The P$_{TetO2}$-TLC1 *hog1Δ* cells, treated with NAC 10 mM during the exponential phase, exhibit an increase in cell viability coupled with a decrease of ROS levels. These results are consistent with a model where Hog1 and Mec1 are both involved in ROS detoxification during replicative senescence but act in independent pathways.

## Discussion

Here, we have shown that increased ROS levels are a feature of replicative senescence in budding yeast. Hog1, one of the five MAPKs of *S. cerevisiae*, is activated by Pbs2 during replicative senescence and counteracts increases in ROS levels, likely independently of Mec1. In addition, Hog1 regulates telomere length homeostasis, and its deletion results in a marked increase in cell mortality rates. Our findings also indicate that autophagic processes are not essential in the context of replicative senescence in budding yeast.

Previous studies have shown that Hog1 is activated in response to exogenous acute stresses, such as H$_2$O$_2$ exposure, where it is essential for triggering antioxidant genes and maintaining cell viability[28,31]. Given our findings that ROS levels increase during replicative senescence, it is plausible that oxidative stress directly triggers Hog1 pathway activation. Replicative senescence is an endogenous process resulting from telomerase inhibition, that leads to numerous cellular modifications at both genomic and

metabolic levels. Consequently, other modifications may also contribute to Hog1 activation. Notably, a previous study proposed that Hog1 activation in response to H$_2$O$_2$ stress primarily occurs through the Sln1-Ypd1-Ssk1-Ssk2-Pbs2 pathway, with Ssk2 acting as the MAPKKK that specifically activates Hog1 in response to oxidative stress, but not Sho1 branch[61]. Therefore, determining the Hog1 pathway upstream of Pbs2 might help clarify the origin of Hog1 pathway activation in the absence of telomerase.

Microfluidics analysis, where cells grow individually, showed that *HOG1* deletion in the presence of telomerase increased cell mortality rates by ~15 fold. This increase may have gone undetected in other studies where cells were grown in populations as colonies or liquid cultures due to competition and selection of the fittest cells. Similar mortality rates have been described for other mutants considered "viable", underscoring the high sensitivity of the microfluidics method[46]. We speculate that the Hog1 pathway might be important in response to certain intrinsic stresses, and that it becomes essential in rare situations. Accordingly, a potential role for Hog1 under normal stress-free cellular conditions, unrelated to telomeres, has been described[62].

We have shown that Hog1 participates in telomere length homeostasis in budding yeast. This could be due to Hog1 acting to positively regulate telomere transcriptional silencing through the localization of the Sir complex following osmotic stress[39]. It could also be that the absence of Hog1 disrupts subtelomeric heterochromatin, which would alter telomere length homeostasis.

In conclusion, this study has shown that the metabolic alterations observed in human senescent cells are conserved in budding yeast. These alterations involve a conserved MAPK Hog1/p38 pathway, although the outcome might differ in different species. As most basic functions in telomere biology are conserved in eukaryotes, determining the mechanistic link between telomere shortening and increases in ROS levels in budding yeast will be essential to clarify how telomeres have evolved in the context of eukaryotic evolution.

## Material and methods

### Yeast strains

All yeast strains used in this study were derived from a W303 background corrected for *RAD5* and *ADE2* (Table 1). Gene deletions were constructed as previously described[63]. *Mec1-21* point mutation was constructed using Crispr-Cas9, as previously described[64]. Strains expressing Rosella constructs from plasmids were constructed as previously described[54]. Primers used are listed in Table 2.

### Liquid senescence experiments

Strains were grown at 30 °C in liquid-rich media (YPD). Cell suspensions were diluted to 0.001 OD$_{600nm}$ with a final concentration of 30 μg/mL of doxycycline *(Sigma-Aldrich #D9891)*, and the OD$_{600nm}$ was measured after 24 h. Cultures were similarly diluted for several days and daily samples were taken for analysis.

### ROS detection

Yeast cultures with an OD$_{600nm}$ of 0.4 were incubated at 30 °C in darkness for one hour in 500 μL of sterile 1X PBS, containing DCF-DA (2',7'-Dichlorofluorescin diacetate) *(Sigma-Aldrich #D6883)* at a final concentration of 10 μM. Samples were washed and then resuspended in 500 μL of 1X PBS. Fluorescence was then analysed by flow cytometry using the settings, 488 (λex)/533 (λem) in Accuri C6 or MACSQuant Analyzer 10. The mean intensity values were then plotted. The mean values of P$_{TetO2}$-TLC1 at day 0 without doxycycline were subtracted.

### DNA extraction

Cells with an OD$_{600nm}$ of 5 were centrifuged for 4 min at 2000 g and washed in 500 μL of sterile distilled water. After centrifugation, 200 μL of lysis buffer (Triton 100 X - 10% SDS sodium dodecyl sulfate - 5 M NaCl – 0.5 M EDTA ethylenediaminetetraacetic acid - 1 M Tris - H$_2$O), 200 μL of 0.45 μm acid-washed glass beads *(Sigma-Aldrich #G8772)*, and 200 μL of phenol: chloroform: isoamyl alcohol solution *(25:24:1, Sigma-Aldrich #77617)* were added to the cell pellet. The tubes were vortexed for 15 min at 4 °C, followed

**Table 2 | Primers used in this study**

| Name | Description/resulting strain | Sequence |
|------|------------------------------|----------|
| oT568 | deletion of *TLC1* (yT) | 5'-GCA ATG GTG ACA TAT AGA TCT CAA GGT TCT CAA TTA AAA GAC CTT CTT TGT AGC TTT TAG TGT GAT TTT TCT GGT TTG AGC GGA TCC CCG GGT TAA TTA A-3' |
| oT1569 | deletion of *TLC1* (yT) | 5'-GAC AAT TAC TAG GAT GTT CTT CTA TTT TTT TAT TTT TAT TTG TAT ATT GTA TAT TCT AAA AAG AAG AAG CCA TTT GGT GGG AAT CGA GCT CGT TTT AAA C-3' |
| oT210 | insertion of $P_{TetO2}$ upstream of *TLC1* (yT787) | 5'- AAT ACG ATT AAG CAA ACG CAA CAG CCA TTG ACA TTT TCA TAG GGT ACC TAT CTT CCT CTC ATA GGC CAC TAG TGG ATC TG-3' |
| oT543 | insertion of $P_{TetO2}$ upstream of *TLC1* (yT787) | 5'-AAA AAA CTT CCT CTT TAG CAA TGG TGA CAT ATA GAT CTC AAG GTT CTC AAT AAA AGA CCG GAT CCC CGG GTT ATT AA -3' |
| oT1550 | deletion of *HOG1* (yT1473) | 5'-GGT AAA TAC TAG ACT CGA AAA AAA GGA ACA AAG GGA AAA CAG GGA AAA CTA CAA CTA TCG TAT ATA ATA CGG ATC CCC GGG TTA ATT AA-3' |
| oT1551 | deletion of *HOG1* (yT1473) | 5'-CCA TAA AAA AAA GAA ACA TCA AAA AGA AGT AAG AAT GAG TGG TTA GGG ACA TTA AAA AAA CAC GTG AAT CGA GCT CGT TTT AAA C-3' |
| oT1252 | deletion of *PBS2* (yT1714) | 5'-ATT ATT ATA TTA AGC AGA TCG AGA CGT TAA TTT CTC AAA GCG GAT CCC CGG GTT AAT TAA-3' |
| oT1253 | deletion of *PBS2* (yT1714) | 5'-TAT ATT CAC GTG CCT GTT TGC TTT TAT TTG GAT ATT AAC GGA ATT CGA GCT CGT TTA AAC-3' |
| oT1415 | *Mec1-21* point mutation (yT1743-yT1744) | 5'-AAA CTA CAG GAT AAT ATC TTG TTT T-3' |
| oT1416 | *Mec1-21* point mutation (yT1743-yT1744) | 5'-AAG ATA TTA TCC TGT AGT TTG GAT CA-3' |
| 370 | deletion of *ATG8* (yT1701- yT1702)[a] | 5'-GAT AAG AGA ATC TAA TAA TTG TAA AGT TGA GAA AAT CAT AAT AAA-3' |
| 371 | deletion of *ATG8* (yT1701- yT1702)[a] | 5'-CGA TTT TAG ATG TTA ACG CTT CAT TTC TTT TCA TAT AAA AGA CTA-3' |
| 333 | deletion of *ATG32* (yT1705- yT1706)[a] | 5'-GTC CTA ATC ACA AAA GCA AAA GCG TAC GCT GCA GGT CGA C-3' |
| 369 | deletion of *ATG32* (yT1705- yT1706)[a] | 5'-AAG TGA GTA GGA ACG TGT ATG TTT GTG TAT ATT GGA AAA AGG TTA-3' |
| oT182 | TeloPCR Y' Fwd | 5'-CTG TAG GGC TAA AGA ACA GGG-3' |
| oT169 | TeloPCR Rev | 5'-GCG GAT CCG GGG GGG GGG G-3' |

[a]Obtained from N. Belgareh-Touzé and O. Ozturk

by the addition of 200 μL of TE (Tris/EDTA) at pH 8. After five minutes of centrifugation at maximum speed, the aqueous phase was transferred to a new tube containing the same volume of isopropanol. After mixing by inversion, the samples were centrifuged for 1 min at maximum speed. The resulting DNA pellet was washed in 500 μL of ethanol. Finally, after centrifuging and drying in a speed vacuum (3 min at 40 °C), the DNA pellet was resuspended in 50 μL of TE and 0.1 μL of RNase A (100 mg/ml) and incubated for 30 min at 37 °C. The quality and integrity of the DNA were checked by agarose gel electrophoresis (1% agarose, 0.5X TBE). The quantity was evaluated using Qubit2 *(Thermofisher)*. Samples were stored at −20 °C.

### SDS-PAGE and western blot
Cells with an $OD_{600nm}$ of 5 were collected by centrifugation. The pellet was lysed in 0.2 M of sodium hydroxide (NaOH) for 10 min on ice. After adding trichloroacetic acid (TCA) at a final concentration of 0.5%, the samples were incubated again on ice for 10 min. After centrifuging for 10 min at maximum speed at 4 °C, the pellet was resuspended in 50 μL of Laemmli 4X buffer and denatured at 95 °C for five minutes. Protein samples were electrophoresed on a 10% denaturing gel (or 7.5% for Rad53 detection) of 37.5:1 Acrylamide: Bis-acrylamide *(Sigma-Aldrich #A3699)*. Proteins were then transferred to a nitrocellulose membrane *(Amersham Protran 0.45 NC, GE HealthCare)* and stained with Ponceau red. The following antibodies were used: *Cell Signaling, #9211* to detect Phospho-Hog1, Santa Cruz, *#sc-165978* to detect total Hog1, *Abcam, #ab104252* to detect both unphosphorylated and phosphorylated forms of Rad53, and the horseradish peroxidase-coupled secondary antibody (HRP) *(Sigma, #A9044 and #A9169)*. The signals were revealed using an electrochemiluminescence reagent *(ClarityWestern ECL, Biorad)* and recorded using the ChemiDoc Imaging System (Biorad).

### Telomere-PCR
This method was adapted from ref. 65. In brief, 40 ng of genomic DNA was denatured at 98 °C for 5 min before the tailing reaction in 20 μL of *New England Biolabs* Buffer 4 (1X), 100 μM of dCTP, and 1U of terminal transferase *(New England Biolabs #M0315L)*. The reaction was incubated at 37 °C for five minutes, followed by 5 min at 94 °C, and then maintained at 4 °C. For the PCR reactions, 5 μL of the polyC-tailed products were used with 1X Taq Mg-free Buffer *(New England Biolabs)*, 500 nM of each primer (Table 2), 200 μM of dNTPs, and 1 U of Taq Standard Polymerase *(New England Biolabs #M0273)* in a final volume of 30 μL. The following PCR program was used: 3 min at 94 °C, followed by 45 cycles of 20 s at 94 °C, 15 s at 57 °C, 20 s at 72 °C, and finally, five minutes at 72 °C. The TeloPCR products were then loaded onto a large 2% agarose gel with 1X TBE buffer and 100 ng/ml of ethidium bromide (BET). A 50 bp molecular weight marker was also loaded *(New England Biolabs #N3236)*. Electrophoresis was then performed at 50 V for 15 h. Visualization and analysis were performed using the ImageLab® software *(Biorad)*.

### Microfluidics analysis
Microfluidics analyses were performed as previously described[46].

### Statistics and reproducibility
All data were analysed with excel and are presented as means ± SD. Comparisons between two groups were performed using a two-tailed Student's *t* test. A $P < 0.05$ was considered to be statistically significant. To ensure reproducibility, a minimum of three replicates were conducted with three independent clones. The figure legend details sample sizes and specifies the replicates.

### Data availability
The source data presented in the main figures can be obtained from Supplementary Data 1, while other data can be obtained from the corresponding author upon reasonable request. The uncropped/unedited western blot and telo-PCR images are included in Supplementary Figs. 2, 9 and 4 respectively.

### Code availability
Scripts used with Matlab 2014a to create graphs from microfluidic analysis results are available in Zenodo, a DOI-minting repository. https://doi.org/10.5281/zenodo.11634847.

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

## Acknowledgements

We wish to thank Mohcen Benmounah for writing the script that simplified the analysis and graph creation for microfluidic analysis. Thank to Naima Belgareh-Touzé and Oznür Ozturk for sharing reagents and technical advice, the Teixeira lab, and the UMR8226 unit members for technical support and fruitful discussions. We also wish to thank Pascale Jolivet, Prisca Berardi, Yann Lustig, Juan Manuel Peralta, Pol Ubeda, Victoria Rojat and Clara Basto for technical help, and Francesc Posas' lab, Miguel Godinho Ferreira, and Zhou Xu for fruitful discussions. During the preparation of this work, we occasionally used ChatGPT and DeepL to improve the readability and language of the manuscript. After using this tool, the authors reviewed and edited the content as needed and take full responsibility for the content of the published article. Work in MTT's laboratory was supported by the CNRS, Sorbonne University, the "Fondation de la Recherche Medicale" ("Equipe FRM EQU202003010428"), by the French National Research Agency (ANR) as part of the "Investissements d'Avenir" Program (LabEx Dynamo ANR-11-LABX-0011-01) and the French National Cancer Institute (INCa_15192). B.Z. is a recipient of fellowship from the «Ministère de l'enseignement supérieur et de la Recherche» (MESR).

## Author contributions

A.B. conceived the study. B.Z. and A.B. performed experiments and analyzed the data. B.Z., M.T.T., and A.B. wrote the manuscript. A.B. and M.T.T. supervised the study. M.T.T. provided funding support. All authors discussed the data, read, and edited the manuscript.

## Competing interests

The authors declare no competing interests.
