## [Peer Review File · Communications Biology]

Reviewers' comments:

Reviewer #1 (Remarks to the Author):

ROS levels are elevated in senescent human fibroblasts. In this manuscript, the authors examine the relationship between ROS, the kinases Hog1 and Mec1, and replicative senescence in *Saccharomyces cerevisiae*. They find that ROS increases during replicative senescence; Hog1 and Mec1 respond independently to limit the increase in ROS; and in telomerase-positive cells, deletion of HOG1 causes a slight decrease in telomere length and a reduction in cell viability.

Major issues:

1. The authors speculate that the accelerated senescence observed for telomerase-negative *hog1Δ* strains could be due to the slightly shorter telomere lengths prior to telomerase inactivation. This hypothesis should be verified by doing an experiment starting with HOG1/*hog1Δ* heterozygous diploids that are also TLC1/*tlc1Δ* or TLC1/PTetO2-TLC1. Tetrads are then dissected to yield HOG1 or *hog1Δ* telomerase-negative haploid progeny, which would have started with identical telomere lengths because they came from the same diploid parent.

Minor issues:

2. Why is there a slight increase in ROS even in telomerase-positive cells (-dox in Fig 1B and TLC1 cells in Fig S1)?
3. On p. 5, the authors mentioned “the emergence of post-senescence survivors”. This should be properly explained for non-experts.
4. On p. 6, the authors write that “Hog1 might interfere with telomerase recruitment or activity.” Given that deletion of HOG1 decreases telomere length, it would be more accurate to say that Hog1 might promote telomerase recruitment or activity.
5. The legend for Figure 4B is incorrect, and a description for Figure 4D is missing.
6. There is insufficient description of Figure S4. How does this assay work? What is observed in the images?
7. Is it possible that growth in the microfluidic device is causing the reduction in viability of *hog1Δ* cells?

Reviewer #2 (Remarks to the Author):

This study examines in budding yeast, provides evidence for Hog1 in suppressing ROS elevation

that occurs after the onset of senescence in telomerase-deficient cells. Furthermore, the authors show that Hog1 deficient cells show shortened telomeres and accelerated senescence. A mutant form of Mec1 that is defective in S phase checkpoint also accelerates senescence and leads to increased ROS when telomerase is inactivated, but shows a synergistic effect with Hog1 loss. Growth of the triple mutant cells is severely impaired and ROS levels are greatly increased. This led the authors to conclude that Hog1 and Mec1 function independent pathways. Overall, the manuscript is well written and the data support the conclusions. I only had a few minor comments.

Page 6 – the last sentence of paragraph 1 requires clarification. The authors show that telomerase-negative budding yeast show increased ROS, but is this due to senescence or telomerase loss? In other words, would budding yeast that maintain telomeres by ALT show increased ROS?

Figure 2C shows a correlation between increased phosphorylation of Hog1 and a reduction in cell growth (A) and an increase in ROS (C). But do senescent cells show Hog1 phosphorylation? Perhaps the authors can infer this from the day 5 data if all the cells stop growing.

Figure 2E-F, does Hog1 loss alone in telomerase positive cells alter mortality rates?

Figure 4C, the defect in RAD53 phosphorylation in the *mec1-21* strain is not readily evident. Please clarify. Also, the legend for 4D is missing. Could the much higher levels of ROS in the triple mutant cells (magenta line and bar) be due to an increase in the fraction of senescent cells rather than a role for Mec1 in ROS detoxification?

Reviewer #3 (Remarks to the Author):

In this paper “Hog1 acts in a Mec1-independent manner to counteract oxidative stress following telomerase inactivation”, Zeinoun et al, show in a model of replicative senescence in *Saccharomyces cerevisiae* induce by inactivation of telomerase a role of Hog1 in ROS production. They hypothesize that HOG1 deletion can impact telomere length in presence of telomerase and impact cell viability. Finally, they show that HOG1 acts from an independent pathway from Mec-1 to regulate ROS level during Yeast replicative senescence.

The paper is in the overall, clear and logical. The conclusions are supported by the data presented. However, the implication of p38 MAPK in the oxidative stress and cellular senescence has already been extensively shown in mammalian cells, so I might questioned a little the relevance of this study in yeast and the novelty of it.

Some point should be addressed to improve the manuscript before publication:

Figure 1:

Replicative senescence is reached at D5 in your model, not D3. Replicative senescence is reached

when there is cell cycle arrest and no more cellular division. Moreover, the ROS level increased at D3 for both of your condition in this figure. And level stay equal for the control after that. I would suggest changing that in the text, to be more precise. Data are however clear for D5 and 6.

Figure S1:

It is not clear which condition you use for these experiments? Can you clarify? How can cells escape cell arrest if they don't have telomerase and are able to growth again? Could you show different WB gene KO of your cell stain to be sure they don't reexpress tlc1? Why two graphs?

Figure 2:

- WB of Hog1 and pbs2 are necessary to show the KO of the protein and confirm the phenotype of the cells.
- B: Can you clarify on the graph that you treat Dox since Day 1 for this experiment? Same for graph C. Is it in basal or after Dox treatment?
- B: What are the ROS levels in pbs2 Δ cells?
- What are the ROS levels in hog1 Δ cell after several replications without dox induction. And their proliferation level? Does hog1 Δ has impact on those on basal conditions.
- Specify somewhere in the text that Δ mean deletion and HOG1 mean normal expression. Will be easier for the reader to follow.
- What will be the cell viability and proliferation of hog1 Δ cell treated with antioxidant or ROS scavenger? This control is essential for your whole study of the implication of ROS on the physiopathology.

Figure 3:

- To help the reader could you show with some line where you quantify the telomere length in all your telomere-PCR.
- Telomere length in not the only explanation for accelerated senescence. DNA damage at telomere has been shown to induce senescence in many mammals cell type and tissues. ROS is also an inducer of DNA damage at telomere. I would recommend to check the DNA damage at telomere (TAF) by staining in your model (<https://doi.org/10.1038/ncomms1708>).
- ROS seem to be increased in basal level D0 by deletion of HOG1. What would be the telomere length in cells hog1 Δ treated by antioxidant or ROS scavenger?
- Figure E and F: Are they treated with Doxo or not? Not clear.

Figure 4:

- WB needed for all mutant to show the KO
- HU expose? What is it?
- "consistent with initial shorter telomere" => Where are the data in telomere length in mec1 Δ

Point by point response to the reviewers' comments on the
Communications Biology submission:
RSID: rs-3528534

We deeply appreciate the opportunity to review our work and are sincerely grateful for the insightful feedback provided by the editor and the reviewers, all of which we have fully addressed. Please see our detailed responses to the reviewers' comments outlined below (in light blue). References to page and figure numbers correspond to those in the final manuscript.

Reviewer #1: genome integrity, telomeres (Remarks to the Author):

Major issues:

1. The authors speculate that the accelerated senescence observed for telomerase-negative *hog1Δ* strains could be due to the slightly shorter telomere lengths prior to telomerase inactivation. This hypothesis should be verified by doing an experiment starting with *HOG1/hog1Δ* heterozygous diploids that are also *TLC1/tlc1Δ* or *TLC1/PTetO2-TLC1*. Tetrads are then dissected to yield *HOG1* or *hog1Δ* telomerase-negative haploid progeny, which would have started with identical telomere lengths because they came from the same diploid parent.

We thank the reviewer for suggesting this experiment, which helped us to separate the slight decrease in telomere size observed in *P_{TetO2}-TLC1 hog1Δ* compared to *P_{TetO2}-TLC1* (new Figure S4A and B, formerly Figure 3A and 3B) prior to telomerase inactivation from the accelerated senescence observed in this strain upon telomerase inactivation.

We proceed to dissection of *P_{TetO2}-TLC1/P_{TetO2}-TLC1 HOG1/hog1* diploids and found substantial variability in telomere length among a majority of clones of same genotype (presented in new supplementary Figure 4C and D). Significant inter-clonal variability in telomere length was previously reported in the study by Shampay *et al.* (1988). In this context, the small but reproducible decrease in telomere size of independent *P_{TetO2}-TLC1 hog1Δ* transformants compared to *P_{TetO2}-TLC1* is no longer significant in a context of tetrad dissection. Yet, the proliferation of *P_{TetO2}-TLC1 hog1Δ* spores in the presence of doxycycline, consistently show accelerated senescence compared to *P_{TetO2}-TLC1 HOG1* sister spores. Based on these new results, we can conclude that the acceleration of senescence in the absence of Hog1 is independent of the initial shorter telomere length observed in this strain.

Nevertheless, we would like to emphasize that to answer to another comment, we repeated the measurements of telomere lengths of independent transformants *P_{TetO2}-TLC1 hog1Δ* and compared to *P_{TetO2}-TLC1 HOG1* and found again a slightly shorter telomere length upon *HOG1* deletion (new Figure S5).

Minor issues:

2. Why is there a slight increase in ROS even in telomerase-positive cells (-dox in Fig 1B and TLC1 cells in Fig S1)?

In Figure 1B and Supplementary Figure 1 (S1), the observed differences do not exhibit statistical significance across various days. Therefore, we conclude that this variability is intrinsic to the system, likely reflecting biological variation.

3. On p. 5, the authors mentioned “the emergence of post-senescence survivors”. This should be properly explained for non-experts.

To properly explain “the emergence of post survivors”, we have included the following paragraph in the manuscript p5:

“As previously outlined, during prolonged culturing of senescent cells, rare events, estimated to occur at a frequency of approximately 2×10^{-5} cells, enable cells to circumvent senescence and resume cellular divisions (Lundblad & Blackburn, 1993, Kockler et al., 2021). These dividing cells, referred to as post-senescence survivors, have the ability to replenish the culture and sustain their telomere length independently of telomerase activity. Instead, they rely on recombination mechanisms, known as the alternative lengthening of telomeres (ALT) mechanism, which are conserved in yeast and mammals.”

4. On p. 6, the authors write that “Hog1 might interfere with telomerase recruitment or activity.” Given that deletion of HOG1 decreases telomere length, it would be more accurate to say that Hog1 might promote telomerase recruitment or activity.

Indeed, thank you for your comment. We have incorporated the changes into the text.

5. The legend for Figure 4B is incorrect, and a description for Figure 4D is missing.

We have made the required adjustments to improve the clarity of the legend of Figure 4B. Moreover, we have included the legend of Figure 4D.

6. There is insufficient description of Figure S4. How does this assay work? What is observed in the images?

To rectify this lack of information, we have incorporated a dedicated "Materials and Methods" section for the experiment in supplementary information of the Figure S6:

Cells harbouring a biosensor Cytosella and Mitrosella consisting of a rapidly maturing, pH-stable red fluorescent protein fused with a pH-sensitive variant of green fluorescent protein. The fluorescence of these proteins depends on variations in pH across various cellular

compartments and the vacuole. The Cytosella and Mitorosella constructions are used to follow autophagy and mitophagy respectively. Cells were diluted to 0,1 OD_{600nm} respectively in SD-Leu and SG-Leu at 30 °C under 220 rpm agitation. When cells reached 0,8 OD, they are washed 3 times with water and transferred to SD-N media (synthetic dextrose media lacking Nitrogen) for 24 hours to induce autophagy or mitophagy and visualized using a fully motorized Axio Observer Z1 inverted microscope (Zeiss) with DsRed, GFP and Phase settings.

Additionally, we have added a more explicit legend to Figure S6:

“Cytosella is a fusion between a cytoplasmic targeting signal fused to DsRed and to a GFP sensitive to pH. Mitorosella is similar to cytosella but with a mitochondrial targeting signal. Images of fluorescence microscopy following autophagy (A) and mitophagy (B) induced by nitrogen depletion for 24 hours. (A) *P_{TetO2}-TLC1* and *P_{TetO2}-TLC1 hog1Δ* strains transformed with Cytosella exhibit red fluorescence accumulation in the vacuole reflecting autophagy process. Lack of green fluorescence, indicates the delivery of Cytosella to the vacuole. Conversely, *P_{TetO2}-TLC1 atg8Δ* and *P_{TetO2}-TLC1 hog1Δ atg8Δ* strains, used as a positive controls, do not show accumulation of either fluorophore in the vacuole. (B) For mitophagy process, similarly, following 24 hours of nitrogen depletion, *P_{TetO2}-TLC1* and *P_{TetO2}-TLC1 hog1Δ* strains transformed with Mitorosella display red fluorescence accumulation within the vacuole while lacking green fluorescence, indicating the delivery of mitochondria to the vacuole. Conversely, *P_{TetO2}-TLC1 atg32Δ* and *P_{TetO2}-TLC1 hog1Δ atg32Δ* strains, used as a positive controls, do not exhibit accumulation of either fluorophore in the vacuole.”

All phenotypic analyses presented in Supplementary Figure 6 confirm the deletion of the *ATG8* and *ATG32* genes from the strains used.

7. Is it possible that growth in the microfluidic device is causing the reduction in viability of *hog1Δ* cells?

Actually, we cannot rule out the possibility that the deletion of *HOG1* affects cell proliferation specifically within the cavity of the microfluidic system. Therefore, we assessed the proliferation in population of the *P_{TetO2}-TLC1 hog1Δ* strain, ensuring that cells remain consistently in the exponential phase. Under these conditions, we were also able to detect the proliferation defect of the *P_{TetO2}-TLC1 hog1Δ* strain compared to the *P_{TetO2}-TLC1* strain. These new data, presented in supplementary Figure 5A, demonstrate that the proliferation defect in *P_{TetO2}-TLC1 hog1Δ* cells is not specific to the microfluidic device.

Reviewer #2: ROS, telomeres, senescence (Remarks to the Author):

Page 6 – the last sentence of paragraph 1 requires clarification. The authors show that telomerase-native budding yeast show increased ROS, but is this due to senescence or telomerase loss? In other words, would budding yeast that maintain telomeres by ALT show increased ROS?

In our experiment, we intentionally inhibited telomerase activity, resulting in several observed effects: 1) telomere shortening, 2) entry into senescence, and 3) other consequences. Therefore, we cannot determine which of these phenomena directly lead to the observed increase in reactive oxygen species (ROS). However, cells using alternative lengthening of telomeres (ALT), characterized by heterogeneous telomeres and a small proportion of senescent cells (as demonstrated by Misino et al. (2023)) within the population, do not exhibit any ROS increase. This suggests that mere telomerase inactivation is not the primary cause of the ROS elevation. However, it remains plausible that the presence of shortened telomeres, a higher proportion of senescent cells, or other consequences resulting from telomerase inactivation might contribute to the observed ROS increase.

Figure 2C shows a correlation between increased phosphorylation of Hog1 and a reduction in cell growth (A) and an increase in ROS (C). But do senescent cells show Hog1 phosphorylation? Perhaps the authors can infer this from the day 5 data if all the cells stop growing.

Indeed, there is a correlation between the loss of proliferation capacity of the population and increase in Hog1-P signal, suggesting that senescent cells may be the ones in which Hog1 is actually activated. However, single analyses would be required to answer this point. Future work to establish this is planned, but it's beyond the scope of the present manuscript.

Figure 2E-F, does Hog1 loss alone in telomerase positive cells alter mortality rates?

The reviewer is referring here to Fig 3E-F. Since doxycycline was omitted from the microfluidic experiment, telomerase remains consistently active throughout the procedure. To clarify this, we included 'without dox' in the illustration. We can therefore conclude that deletion of *HOG1* alone in telomerase-positive cells leads to a 15-fold increase in mortality compared to the reference strain.

Figure 4C:

- The defect in RAD53 phosphorylation in the *mec1-21* strain is not readily evident. Please clarify.

We have incorporated additional arrows indicating the signal corresponding to the unphosphorylated form of Rad53, and the various phosphorylated forms of Rad53, which result

in a smeary signal above the unphosphorylated Rad53. In Figure 4C, we can observe that the smeary signal is consistently absent in the *P_{TetO2}-TLC1 mec1-21* strain throughout the senescence experiment, indicating a deficiency in checkpoint activation.

- Also, the legend for 4D is missing.

We are sorry for this mistake. We have added the legend for Figure 4D.

- Could the much higher levels of ROS in the triple mutant cells (magenta line and bar) be due to an increase in the fraction of senescent cells rather than a role for Mec1 in ROS detoxification?

Actually, both interpretations are possible. In Figure 4B, we observed a slight decrease in proliferation in the single *P_{TetO2}-TLC1 mec1-21* strain even prior to telomerase inactivation (at D0), indicating an increase in the proportion of deceased cells. This trend is further pronounced in the double mutant *P_{TetO2}-TLC1 mec1-21 hog1*. Therefore, it is well possible that ROS levels follow mortality. This observation remains consistent for the ROS levels observed in Figure 4D at D0. However, a role of Mec1 in maintaining ROS homeostasis is established in the 2014 paper by Tsang *et al.*. Additional experiments have been incorporated into Supplementary Figure 5A and B. The *P_{TetO2}-TLC1 hog1Δ* cells, treated with 10 mM of the antioxidant N-acetyl-L-cysteine (NAC) during the exponential phase, exhibit an increase in cell viability coupled with a decrease of ROS levels. We thus favor a model in which Hog1 and Mec1 cooperate to maintain ROS homeostasis. To clarify this point, we modified the paragraph p9.

Reviewer #3: oxidative stress, cellular senescence (Remarks to the Author):

Some point should be addressed to improve the manuscript before publication:

Figure 1:

Replicative senescence is reached at D5 in your model, not D3. Replicative senescence is reached when there is cell cycle arrest and no more cellular division. Moreover, the ROS level increased at D3 for both of your condition in this figure. And level stay equal for the control after that. I would suggest changing that in the text, to be more precise. Data are however clear for D5 and 6.

Thank you for your comment, on p5 we have modified the text accordingly.

Figure S1:

It is not clear which condition you use for these experiments? Can you clarify? Why two graphs?

The experiment presented in Supplementary Figure 1 is based on haploid spores obtained through tetrad dissection of a *TLC1/tlc1Δ* diploid. This approach allows us to monitor the division of the four haploid strains stemming from the same diploid segregating 2:2. Each genotype is represented by two graphs, corresponding to the two haploid strains with the *TLC1* genotype and the two haploid strains with the *tlc1Δ* genotype, obtained from the dissection of a single tetrad. We clarified this procedure in the legend of Supplementary Figure 1.

How can cells escape cell arrest if they don't have telomerase and are able to growth again?

We included the following paragraph on p5 to provide a comprehensive explanation of post-senescence survivors:

“As previously outlined, during prolonged culturing of senescent cells, a rare event, estimated to occur at a frequency of approximately 2×10^{-5} cells, enables cells to circumvent senescence and resume cellular divisions (Kockeler *et al.*, 2021; Lundblad & Blackburn, 1993). These dividing cells, referred to as post-survivors, have the ability to replenish the culture and sustain their telomere length independently of telomerase activity. Instead, they rely on recombination mechanisms, known as the alternative lengthening of telomeres (ALT) mechanism, which are conserved in yeast and mammals.”

Could you show different WB gene KO of your cell stain to be sure they don't reexpress *tlc1*?

For all *TLC1* deletion mutants examined in this study, we replaced the *TLC1* gene, encoding telomerase template RNA, a non-coding RNA, with a cassette providing resistance to the antibiotic nourseothricin (Nat) by homologous recombination, which is very efficient in budding yeast and enables accurate gene editing. Genome experimental modifications are then checked by PCR analyses. We further test the correct deletion by phenotypic analyses. Upon tetrad dissection, we verify 2:2 co-segregation of Nat resistance with the gradual loss of cell proliferation, typical of *TLC1* loss as previously described in Singer and Gottschling paper (1994).

Figure 2:

- WB of Hog1 and *pbs2* are necessary to show the KO of the protein and confirm the phenotype of the cells.

In *Saccharomyces cerevisiae*, antibodies are not available for many proteins, and we mainly rely on phenotypic verification of genotypes, as indicated above. Yet, when available, we do this verification. We have thus added the Western blot showing the deletion of *HOG1* in Figure 2C. For *Pbs2*, antibodies are not available. However, *Pbs2* is the only kinase capable of phosphorylating Hog1. Therefore, the Western blot presented in Figure 2D, showing that the phosphorylated form of Hog1 is no longer detectable in the *PBS2*-deleted mutant, confirms the deletion.

- B: Can you clarify on the graph that you treat Dox since Day 1 for this experiment? Same for graph C. Is it in basal or after Dox treatment?

We have added to the legend of Figure 2B and 2C that the treatments were identical to those described in Figure 2A. Thus, doxycycline is added only from day 1 onwards (Day 1 is included).

- What are the ROS levels in *pbs2*Δ cells?

We have added this new data to Supplementary Figure 2B. The levels of ROS observed in the *PBS2* deletion mutant do not differ from those observed in the *HOG1* deletion mutant. Since *Pbs2* is the only kinase capable of phosphorylating Hog1 at these two sites, these results confirm the epistasis of the two genes.

- What are the ROS levels in *hog1Δ* cell after several replications without dox induction. And their proliferation level? Does *hog1Δ* has impact on those on basal conditions.

Thanks to your comment, we assessed cell proliferation in the population using the *P_{TetO2}-TLC1* and *P_{TetO2}-TLC1 hog1Δ* strains, ensuring that cells grew in the exponential phase all along the experiment. Under these conditions, after 10 generations, we were able to detect a proliferation defect (Figure S5A) and a constant reproducible higher level of ROS (Figure S5B) of the *P_{TetO2}-TLC1 hog1Δ* strain compared to the *P_{TetO2}-TLC1* strain. These high levels of ROS are also detected when cells are grown to saturation, as shown in Figure 2B and 4D at D0.

- Specify somewhere in the text that Δ mean deletion and HOG1 mean normal expression. Will be easier for the reader to follow.

We have specified this on page 6 and in the legend of the supplementary Figure 4C.

- What will be the cell viability and proliferation of *hog1Δ* cell treated with antioxidant or ROS scavenger? This control is essential for your whole study of the implication of ROS on the physiopathology.

Thank you for asking this. Additional experiments have been incorporated into Supplementary Figure 5A and B. The *P_{TetO2}-TLC1 hog1Δ* cells, treated with 10 mM of the antioxidant N-acetyl-L-cysteine (NAC) during the exponential phase, exhibit an increase in cell viability coupled with a decrease of ROS levels. This further underscores the pivotal role of Hog1 in regulating ROS levels.

Figure 3:

- To help the reader could you show with some line where you quantify the telomere length in all your telomere-PCR.

To improve the understanding of telomere size quantifications, we have added a line representing the peak of the signal using ImageLab (BioRad) software for all telomere-PCR gels.

- Telomere length is not the only explanation for accelerated senescence. DNA damage at telomere has been shown to induce senescence in many mammalian cell type and tissues. ROS is also an inducer of DNA damage at telomere. I would recommend to check the DNA damage at telomere (TAF) by staining in your model (<https://doi.org/10.1038/ncomms1708>).

Indeed, it would be extremely interesting to be able to detect and quantify repair foci at telomeres in yeast cells. Unfortunately, to our knowledge, this technique has not been adapted to yeast nuclei due to their small size and less condensation of chromosomes in metaphase.

- ROS seem to be increased in basal level D0 by deletion of HOG1. What would be the telomere length in cells *hog1Δ* treated by antioxidant or ROS scavenger?

In the new supplementary figure 5C and D, the *P_{TetO2}-TLC1 hog1Δ* cells treated in exponential phase with 10 mM NAC show a partial rescue of telomere length confirming data showing that telomeres are highly sensitive to ROS due to their G-rich sequence (Barnes et al., 2019).

- Figure E and F: Are they treated with Dox or not? Not clear.

We apologize for any confusion caused. The microfluidic experiment was conducted in the absence of doxycycline, therefore in the presence of telomerase. We have clarified this on both the figure and the legend of figures 3E and 3F.

Figure 4:

- WB needed for all mutant to show the KO
- HU expose? What is it?

As explained above, antibodies are not available for many yeast proteins. *ATG8* and *ATG32* deletions were thus confirmed by phenotypic analyses described in Figure S6. Similarly, antibodies are not available for *mec1-21*. The *mec1-21* point mutation was introduced using a Cas9 directed against *MEC1* genomic sequence and a donor DNA containing a repair template containing the point mutation, according to a published method (Lemos *et al.*, 2018). The resulting *mec1-21* locus of independent transformants was sequenced and the *mec1-21* phenotype was confirmed through lack of Mec1-dependent phosphorylation of Rad53 during senescence (Figure 4C).

- “consistent with initial shorter telomere” => Where are the data in telomere length in *mec1Δ*

In the paper Richie *et al.*, 1999, it is described that *mec1-21* cells display short telomeres. Our study mainly uses a *P_{TetO2}-TLC1* allele, in which telomerase is less expressed due to the change in promoter, even in the absence of doxycycline. We thus performed telomere-PCR to determine the telomere length of *P_{TetO2}-TLC1 mec1-21* cells. The results presented in the supplementary Figure 7 show significantly shorter telomeres in the mutant *P_{TetO2}-TLC1 mec1-21* compared to the reference strain *P_{TetO2}-TLC1*. This suggests that *mec1-21* similarly shortens telomeres in both *TLC1* and *P_{TetO2}-TLC1* backgrounds.

REVIEWERS' COMMENTS:

Reviewer #1 (Remarks to the Author):

I am largely satisfied with the modifications made to the manuscript. However, the authors should double check that the references to figures in the text are correct. For example, on page 7, Figures 3E and 3F are mentioned (these figures don't exist). And on the top of page 8, a reference to Figure S4C-D should be Figure S5C-D.

Reviewer #2 (Remarks to the Author):

The authors have satisfied my prior concerns.

Reviewer #3 (Remarks to the Author):

In this second submission of the following paper "Hog1 acts in a Mec1-independent manner to counteract oxidative stress following telomerase inactivation", authors have answered and clarified all the point that I have raised in my first review. I do not have more suggestions at that time. I will just add the following: For more clarity, this following given explanation about the observation of PBS2 deletion could be added in the manuscript. "However, Pbs2 is the only kinase capable of phosphorylating Hog1. Therefore, the Western blot presented in Figure 2D, showing that the phosphorylated form of Hog1 is no longer detectable in the PBS2-deleted mutant, confirms the deletion." Otherwise, I am satisfied by the answers and the work added by the authors.